# Gene Cloning and Characterization of Transcription Factor *FtNAC10* in Tartary Buckwheat (*Fagopyrum tataricum* (L.) Gaertn.)

**DOI:** 10.3390/ijms242216317

**Published:** 2023-11-14

**Authors:** Jinghuan Li, Xiaohua Li, Caihua Jia, Dahui Liu

**Affiliations:** 1School of Pharmacy, Hubei University of Chinese Medicine, Wuhan 430070, China; jane1042@163.com (J.L.); liudahui@hbtcm.edu.cn (D.L.); 2Beijing Key Laboratory of Maize DNA Fingerprinting and Molecular Breeding, Maize Research Institute, Beijing Academy of Agriculture and Forestry Sciences, Beijing 100097, China; 3Department of Life Science and Technology, Huazhong Agricultural University, Wuhan 430070, China; 4Key Laboratory of Environment Correlative Dietology (Ministry of Education), College of Food Science and Technology, Huazhong Agricultural University, Wuhan 430070, China; chjia@mail.hzau.edu.cn

**Keywords:** Tartary buckwheat, *FtNAC10*, transcription factors, stress

## Abstract

NAC transcription factors play a significant role in plant stress responses. In this study, an NAC transcription factor, with a CDS of 792 bp encoding 263 amino acids, was cloned from *Fagopyrum tataricum* (L.) Gaertn. (*F. tataricum*), a minor cereal crop, which is rich in flavonoids and highly stress resistant. The transcription factor was named FtNAC10 (NCBI accession number: MK614506.1) and characterized as a member of the NAP subgroup of NAC transcriptions factors. The gene exhibited a highly conserved N-terminal, encoding about 150 amino acids, and a highly specific C-terminal. The resulting protein was revealed to be hydrophilic, with strong transcriptional activation activity. FtNAC10 expression occurred in various *F. tataricum* tissues, most noticeably in the root, and was regulated differently under various stress treatments. The over-expression of FtNAC10 in transgenic *Arabidopsis thaliana* (*A. thaliana*) seeds inhibited germination, and the presence of FtNAC10 enhanced root elongation under saline and drought stress. According to phylogenetic analysis and previous reports, our experiments indicate that FtNAC10 may regulate the stress response or development of *F. tataricum* through ABA-signaling pathway, although the mechanism is not yet known. This study provides a reference for further analysis of the regulatory function of FtNAC10 and the mechanism that underlies stress responses in Tartary buckwheat.

## 1. Introduction

NAC transcription factors (TFs) form the fourth largest TF family in plants and are involved in regulating plant growth and development, as well as their responses to external stimuli [1,2,3,4]. NAC TFs are named according to the NAM, ATAF1 and CUC1/CUC2 TF genes [5,6]. The NAM gene was first identified in Petunia plants without apical meristem, and CUC1/CUC2 was found to have a similar function in *A. thaliana* [7]. There are known to be 117 NAC TFs in *A. thaliana*, 151 in Oryza sativa, 152 in Nicotiana tabacum and 152 in soybean [8,9,10]. Most NAC proteins have a highly conserved DNA-binding domain at the N-terminal and transcription activation domains at the C-terminal [11].

A number of studies have shown that NAC TF genes are involved in plant adaptations to abiotic stresses such as drought, salt, cold conditions and high temperatures during growth and development [12,13,14,15]. Two NAC genes, *AtAF1* and *AtAF2*, are induced during the response to drought and wounding in *A. thaliana* [16,17,18]. Others, including *AtNAC072* (RD26), *AtNAC019* and *AtNAC055*, are involved in biological and abiotic stress responses and lignin biosynthesis [19,20,21]. Functional studies of NAC TFs have been reported in many plant species. It is indicated that overexpression of *ONAC066* significantly enhanced tolerance to drought and oxidative stress in rice (*Oryza sativa*) [22]. *OsNAC2* not only negatively regulates root growth, but also affects the height of rice [23,24]. *OsNAC3* positively regulates ABA biosynthesis and enhances tolerance to salt stress [25]. *LcNAC002* positively regulates chlorophyll degradation and anthocyanin synthesis by co-activating the expression of *LcSGR* and *LcMYB1* in litchi (*Litchi chinensis* Sonn.) [26], beside, *LcSGR* might promote ROS and ABA biosynthesis by accelerating chlorophyll degradation and chloroplast aging, and indirectly promote *LcMYB1* expression and anthocyanin accumulation. Overexpression of *PsnNAC090*, which was isolated from *Populus simonii* × *Populus nigra,* in transgenic tobacco shows stronger tolerance to salt stress [27].

Tatary buckwheat (*Fagopyrum tataricum*) is an annual dicot that mainly cultivated in China, Russia and India and is grown as a cereal crop [28,29]. In China, *F. tataricum* is planted widely in alpine mountain regions and remote areas such as Yunnan, Sichuan and Guizhou due to its extreme resistance to drought [30]. This crop is grown as one of the eight major cereals, as it is rich in lipids, vitamins (B1, B2 and B6), dietary fiber and minerals, with a protein content of 20–25% and a well-balanced amino acid composition [31]. *F. tataricum* has, therefore, become increasingly recognized and consumed as a functional food with high nutritional value [32]. With respect to its tolerance to biotic and abiotic stress, transcriptome data analysis has indicated that the expression of MYB, NAC, WRKY, bZIP and bHLH TFs is likely to be involved in regulating the adaptations of *F. tataricum* to environmental pressures [30]. Some published data show that FtANC genes are involved in *Tatary buckwheat* biological stress, abiotic stress response mechanism, and regulate plant growth and development. Recently, there are two genes, *FtNAC16* and *FtNAC31,* have been reported, they have shown the important role in response to abiotic stress [33,34]. In addition, *FtNAC16* promotes the biosynthesis of lignin and root elongation [33]. In this research, an NAC transcription factor *FtNAC10* was cloned from *F. tataricum*, the basic function of *FtNAC10* was investigated in order to provides a theoretical basis for further analysis of the regulatory function of *FtNACs* and the mechanism that underlies stress responses in *F. tataricum*.

## 2. Results

### 2.1. Profile of FtNAC10 Expression in Different Tissues of F. tataricum

To study the expression pattern of FtNAC10 in different tissues of *F. tataricum*, we performed quantitative real-time (qRT)-PCR and noted that the FtNAC10 was expressed at levels that were 50-fold greater than those seen in root than leaf tissues (Figure 1). Moreover, expression was about 11 times higher in pre-filling stage seeds (S1) than leaves. In fact, the lowest level of FtNAC10 expression was observed in leaves.

### 2.2. Expression of FtNAC10 under Different Stresses

To study the functional responses of *FtNAC10* to different stresses, we applied abscisic acid (ABA), methyl jasmonate (MeJA), salt (NaCl) and salicylic acid (SA) to *F. tataricum* plants to simulate stress conditions. The results (Figure 2) showed that the expression of *FtNAC10* was up-regulated under ABA, MeJA and SA stress. In particular, the TF was expressed at 12.6-fold higher levels after 24 h of MeJA stress, compared with the control group. Under ABA stress, the expression of FtNAC10 initially increased and then decreased, and reached to the peak at 72 h of treatment, when levels were about 2.2 times higher than those seen in the control group. After 12, 24 and 48 h of salt stress, the expression of *FtNAC10* was down-regulated; however, it was then up-regulated after 72 and 96 h of exposure to NaCl.

### 2.3. Cloning and Characterization of FtNAC10

FtNAC10 was isolated and cloned from *F. tataricum* and found to contain a complete open reading frame (ORF) of 792 bp. The sequence was submitted to NCBI with GenBank accession number MK614506. A BLASTX analysis indicated that FtNAC10 is highly homologous to AtNAC2 (AAM63330.1), with 68.42% identity, and AtNAP (NP_564966.1), with 67.98% identity. FtNAC10 was revealed to encode a protein containing 263 amino acids, with an isoelectric point of 8.38 and a molecular weight of 30,286.21 kDa. Moreover, multiple sequences alignment analysis indicated that, similar to most NAC TFs, FtNAC10 contains relatively highly conserved domains at the N-terminus, which are divided into five sub-domains (A–E), and is highly specific at the C-terminus (Figure 3a).

Next, data about NAC proteins from other plants were used to construct a phylogenetic tree. The results from our analysis of the phylogenetic tree indicated that FtNAC10 belongs to the NAP subgroup of NAC TFs and is most closely related to FtNAC45, FtNAC32, AtNAP and SlNAP2, which are regulatory factors involved in leaf senescence (Figure 3b) [35,36].

### 2.4. Transcriptional Activity of FtNAC10 and Localization of the Protein

To analyze the subcellular localization of FtNAC10, PEGAD-FtNAC10-GFP and PEGAD-GFP were expressed in *N. benthamiana* leaves. The images were observed and compared by confocal microscope with different mode. The fluorescence signal from epidermal cells transformed with PEGAD-GFP was seen in the cytoplasm and nucleus, while the signal from cells transformed with PEGAD-FtNAC10-GFP appeared only in the nucleus (Figure 4a). These results indicated that FtNAC10 is localized in the nucleus.

During the transcription activity assay, the negative control grew on a SD/-Trp plate, but not on a SD/-Trp/His plate. However, the positive control containing the FtNAC10 gene grew well on both the SD/-Trp and SD/-Trp/His plates. Furthermore, both the positive control and pBridge-FtNAC10 showed evident blue color, whereas the negative control showed no color change (Figure 4b,c). *β*-galactosidase activity assay showed that the active unit of β-galactosidase in positive control, pGAL4, was 63.06 U, and the FtNAC10 was 61.22U, with almost the same activity unit (Figure 4d). Overall, the results from the analysis of *β*-galactosidase activity indicated strong FtNAC10 transcriptional activity.

### 2.5. FtNAC10-Associated Inhibition of Transgenic A. thaliana Seed Germination

The relative gene expression of *FtNAC10* in the T2 transgenic *A. thaliana* lines was detected by qRT-PCR (Figure 5). We then chose three lines with higher expression levels, FtNAC10-16 (L16), FtNAC-2 (L2) and FtNAC-10 (L10), for further study. Among them, L16 exhibited the highest level of *FtNAC10* expression, while a lower level of expression was seen in L2 and the lowest in L10. The seed germination assay revealed that the rate of wild-type (WT) seed germination was close to that of L10, while seed germination in L16 was about in line with that of L2, which was lower than WT and L10. During germination, transgenic lines showed lower germination rate at 2 days under normal condition, saline, and drought stress, especially L2 and L16 showed much lower rate than WT (Appendix A). Overall, the germination rates for the three transgenic lines were relatively lower than that of the WT. The germination rates of the WT and L2, L10 and L16 lines after 7 days were 96.6%, 85.2%, 95.9% and 85.4% under normal conditions, respectively. Under saline conditions, these rates were 98.7%, 49.8%, 43.9% and 65.0%, respectively, while under drought stress, they were 96.4%, 50.8%, 91.6% and 40.5%, respectively (Figure 6). Under ABA treatment, the seed germination exhibited a hypersensitive similar trend with other treatments.

### 2.6. Overexpression of FtNAC10 Enhances Salinity and Drought Tolerance in Transgenic A. thaliana

To investigate the effects of FtNAC10 in response to various stresses, 10-day-old transgenic *A. thaliana* seedlings were subjected to saline and drought stress treatments. Our analysis revealed that the primary roots of transgenic *FtNAC10* lines were significantly longer than those of the WT plants under saline and drought stress (Figure 7). In addition, the phenotype of the transgenic lines manifested more lateral roots than the WT under saline stress, and these leaves shrinked less than those of the WT under saline and drought stress. In summary, over-expression of FtNAC10 enhanced the tolerance of transgenic *A. thaliana* in response to saline and drought stress.

## 3. Discussion

According to previously reports, several transcriptional factor family, including MYB, NAC, WRKY, and bZIP, are critically involved in the regulation of plant growth and stress response [37,38,39,40]. It has been reported that NAC transcription factors (TFs), being one of the largest TF families, are involved in fruits development and ripening, stress response, embryonic development, and phytohormone signaling [41]. In previous studies, over-expression of OsNAC5 could enlarge root significantly and enhance the tolerance to drought [42]. Over-expression of OsNAP could enhance the tolerance to abiotic stress and improve yield through the ABA pathway [43]. ANAC032 is involved in root elongation via MYB30 regulation network [44]. To date, research on NAC TFs has focused mainly on model plants such as *A. thaliana*, maize and rice, and the functions of these TFs in these species have been characterized and reported. However, only a few NAC TFs from *F. tataricum* have been described. According to reports, FtNAC45 has the higher expression level in root, leaf and flower, and may play a role in growth and development in *F. tataricum* [31]. A recent study suggested that FtNAC31 plays an indispensable role in the response to drought and salt stress in an ABA signaling-dependent way in *F. tataricum* [34]. Another study revealed that FtNAC16 promotes the biosynthesis of lignin and root elongation under salt stress, which could be of benefit in buckwheat [33].

In this study, we identified an NAC TF, *FtNAC10*, in *F. tataricum*. Analysis of genetic evolution using a phylogenetic tree indicated that *FtNAC10* belongs to the NAP subgroup of NAC TFs and is most closely related to FtNAC45, FtNAC32, AtNAP and SlNAP2. Plant transcriptional factors in the same family usually have similar functions. NAP TFs are associated with regulating plant growth, development and leaf senescence [45]. It has been reported that FtNAC45 may have a role in the growth and development of *F. tataricum* [31]. AtNAP has been reported to bind to the promoter of ABA synthesis gene AOO3 to activate its expression to accelerate leaf senescence eventually [35], while SlNAP2 has been confirmed to be involved in regulating leaf senescence and fruit yield in tomato [36]. Therefore, we inferred that FtNAC10 may play similar roles with FtNAC45, FtNAC32, AtNAP, SlNAP1 and SlNAP2 in plant functional regulation, and FtNAC10 might be associated with fruit development and leaf senescence in *F. tataricum*. The molecular characterization analysis indicated that *FtNAC10* has a highly conserved N-terminal and highly specific C-terminal. Subcellular localization showed FtNAC10 is localized in the nucleus and shows strong transcriptional activation activity. Moreover, the expression profile indicated that *FtNAC10* is expressed most highly in root, less in pre-filling stage seeds and at the lowest level in leaves. Moreover, the expression level of FtNAC10 exhibited an initial increase, followed by a subsequent decrease, and ultimately another increase throughout fruit development, which displayed the similar expression trend with FtNAC45 and may play a role in fruit development and ripening [31]. Applying stressors such as ABA, MeJA, NaCl and SA to plants up-regulated *FtNAC10* expression in *F. tataricum*. This indicated that *FtNAC10* enables these plants to respond to harsh environmental conditions. The seed germination experiments revealed that over-expression of *FtNAC10* in transgenic *A. thaliana* plants inhibited seed germination under normal or stressful conditions, including saline, drought and ABA pressures. And the transgenic L2 and L16 showed significant differences at 2 days or 4 days during generation (Appendix A). we speculate that the expression of FtNAC10 in T3 generation seeds is inadequate, which leads to insufficient sensitivity to these exogenous treatments during *Arabidopsis* seed germination. The expression level of FtNAC10 in L16 was higher compared to L2 and L10 sequentially. Moreover, the seed germination rate of L16 was exactly opposite under normal conditions. We speculated that over-expression of FtNAC10 might directly regulate seed germination via some mechanisms. After 10 days of salinity stress, the root length of L2, L10 and L16 were 0.57 cm, 1.15 cm, 1.48 cm longer respectively, compared to the wild type (WT). Under drought stress, they exhibited an increase in length ranging from 0.08 to 1.3 cm compared to the WT. However, there was no significant difference in root length under normal condition. The seedling stress experiments showed that over-expression of *FtNAC10* promoted root elongation under saline and drought stress in transgenic *A. thaliana*. It suggested that *FtNAC10* might enhance saline and drought tolerance in transgenic *A. thaliana*.

Gibberellin acid (GA) signaling pathway and ABA signaling pathway have been extensively reported to regulate seed germination. The rice transcription repressor, OsWRKY50, plays a role in mediating ABA-dependent seed germination and seedling growth, as well as enhancing salt stress tolerance through an ABA-independent pathway [46]. Overexpression of TERF1 could promote seed germination through glucose signaling pathway, which is mediated by GA [47]. As an important plant hormone, ABA can not only induce resistance to stress in crops but also improve yield, quality, even seed dormancy and seed germination [46]. Numerous studies have found that ABA-signaling pathway plays a vital role in plant responses to stresses such as drought, high salt conditions, low temperatures and invasion by pests or pathogens [12,25,42,44,48,49]. Under stressful conditions, ABA synthesis is activated in plants, resulting in the abundant production of this hormone. This promotes stomatal closure, inhibits stomatal opening, promotes moisture absorption and reduces moisture loss. Previous studies have revealed that the synthesis of ABA is regulated by key enzyme genes such as ABI3, ABI4, RD29A, RD29B, AAO3 and NCED [50,51]. For example, in maize, *ZmWRKY79* indirectly up-regulates the expression of *ZmAAO3* to increase the amount of ABA synthesis and thus enhance drought resistance [52]. In *F. tataricum*, *FtNAC16* may affect tolerance to salinity through a FtNAC16-ABI4-NECD3-ABA-FtNAC16 feedback mechanism [33]. Taken together, we inferred FtNAC10 might be involved with ABA response and positively regulate the synthesis of ABA to inhibit seed germination, like AtNAP, WRKY50, FtNAC16, etc. Whether it has a positive regulatory effect and how it works still need to verify further experiments. Overall, our findings suggest that *FtNAC10* enhances saline and drought tolerance in *F. tataricum*. However, further studies are needed to reveal the specific stress-resistance mechanism mediated by *FtNAC10* in order to improve breeding strategies for this crop in the future.

## 4. Materials and Methods

### 4.1. Plant Materials

The variety used in this experiment was *F. tataricum* 9940, and the seeds were provided by Shanxi academy of agricultural sciences. Seeds were germinated and cultivated in greenhouse under 16 h light/8 h dark cycle at 24 °C condition. And the relative humidity in this room is about 70%. To analyze the expression patterns in different tissues of Tatary Buckwheat, various tissues were collected such as leaves (TB-L), stems (TB-S), roots (TB-R), pre-filling stage seeds (TB-S1), filling stage seeds (TB-S2), fully mature seeds (TB-S3), flower buds (TB-F1), blooming flowers (TB-F2) and near withering flowers (TB-F3) growing in normal condition [53]. These samples were rapidly frozen in liquid nitrogen and then stored at −80 °C for RNA extraction.

### 4.2. Abiotic Stresses Experiments

The seedlings, when growing to One-week-old, were treated with different solutions respectively, such as 1/2 MS containing 100 µM NaCl, 100 µM ABA, 100 µM MeJA, and 100 µM SA. And one pot seedlings were treated with equal 1/2MS containing nothing as wide type (WT). Then all the samples were collected at 0 h, 12 h, 24 h, 48 h, 72 h and 96 h, frozen in the liquid nitrogen at once, and stored at −80 °C for total RNA extraction and qPCR assays. The primer sequences for qRT-PCR were as followed: CCCTTCTACTATCATCCCT (Forward) and ATCAGTTCCTGTGGCTTT (Reverse). Actin was used as reference gene, and each sample was in triplicate. The total reaction system was 10 µL: 5 µL of 2XSYBrR-Gree, 2.5 µL of cDNA, 0.25 µL of each primer, and 2 µL of ddH_2_O. This reaction was performed as following conditions: pre-denaturation at 95 °C for 10 min, 40 cycles of amplification at 95 °C for 10 s, annealing at 54 °C for 30 s and 72 °C for 32 s, melt at 60 °C for 1 min. Gene expression was calculated by 2^−△△Ct^ method [54].

### 4.3. Sequence Analysis and Gene Cloning of FtNAC10

The full-length CDs fragment of FtNAC10 was screened from the transcriptome of *F. tataricum* under salinity stress, and the full-length gene was cloned using the primer designed according to the CDs. The phylogenetic tree analysis between FtNAC10 and some homologous NAC genes or characterized in other species was performed using the Neighbor-Joining (NJ) algorithm in MEGA 5.0 software [9]. The protein sequence of *FtNAC10* was aligned with its homologous NAC genes from other species by Bioedit.

### 4.4. Subcellular Localization and Transcription Activity Assay of FtNAC10

To investigate its exact localization, the ORF of *FtNAC10*, without termination codon, was inserted into PEGAD-GFP vector to generate PEGAD-FtNAC10-GFP plasmid. Agrobacterium GV3101 containing PEGAD-GFP and PEGAD-FtNAC10-GFP were injected into the back of *N. benthamiana* leaves, which were about 4-weeks-old. After dark culture for 48 h, the fluorescence signal of GFP was observed by confocal microscope.

To detect the transcription activity of FtNAC10, the full CDS of FtNAC10 was amplified by PCR with the following primers: 5′-CGGAATTCATGGATCCAAACACAGCC-3′ and 5′-CCAAGCTTTCATGGAAATTTGTAAGC-3′. Then, it was inserted into pBridge vector to obtain the pBridge-FtNAC10 vector. The pBridge (negative control), PGAL4 (positive control) and pBridge-FtNAC10 were transformed into yeast strains AH109, and they were cultured on SD/-Trp-His mediums at 30 °C. Then, detect the transcription activity by *β*-galactosidase colony-lift filter assay according to Yeast Protocol Handbook (Clontech Laboratories, Inc., Mountain View, CA, USA) [55].

### 4.5. Transformation Arabiodopsis

The full CDS was inserted into expression vector PK7FWG2 by Gateway-BP and LR reaction systems to obtain PK7FWG2-FtNAC10. The plasmid PK7FWG2-FtNAC10 was transformed into Agrobacterium tumefaciens GV3101 and then transformation effected Arabiodopsis by dipping flowers method [56]. The transgenic seeds were screened on 1/2 MS medium with 25 µg/mL kanamycin. Positive T3 homozygous lines were confirmed through qRT-PCR and were used for subsequent analysis.

### 4.6. Seeds Germination Experiment

For the statistical analysis of seed germination of transgenic *A. thaliana* under salinity, drought and ABA stress treatments, more than 30 sterilized seeds were placed in 1/2 MS plate medium with 150 mM NaCl, 300 mM mannitol and 0.3 µM ABA, respectively. Seeds were cultured under a 16 h light/8 h dark cycle at 22 °C. The number of seed germinations was counted for 7 days, and photos were taken after sowing 7 days [57].

### 4.7. Seedlings Stress Experiment of Transgenic A. thaliana

*A. thaliana* of each transgenic line was sowed on 1/2 MS medium, vernalized for 3 d at 4 °C, and cultured in an incubator for 10 days, and then transferred to 1/2 MS, 1/2 MS + 150 mM NaCl, 1/2 MS + 300 mM mannitol medium, respectively. Three seedlings of the same strain were transferred to each plate, and there were three plate replicates for each line. After 14 days of upright culture in the incubator, observe the phenotype and root length of *A. thaliana*.

### 4.8. Statistical Analysis

The results all shown in the figures were mean values with standard deviations (±SD) from triplicates. And the data were analyzed by SPSS software (Version 22, SPSS Inc., Chicago, IL, USA). The significance differences were calculated by Duncan’s multiple-range test at a level of 0.05 (*p* < 0.05).

## 5. Conclusions

*FtNAC10* was found to be an important transcription factor, which enhances the sensitivity of *A. thaliana* seeds to salt, drought and ABA stress during germination and inhibits seed germination. In addition, *FtNAC10* was revealed to promote root elongation under saline and drought stress in transgenic *Arabidopsis*. Taken together, the findings suggest that *FtNAC10* may regulate the stress response or development of *F. tataricum* via ABA-signaling pathway. However, further research is required to investigate the mechanism that underlies the *FtNAC10*-associated regulation of ABA synthesis and stress responses in *F. tataricum*.

## Figures and Tables

**Figure 1 ijms-24-16317-f001:**
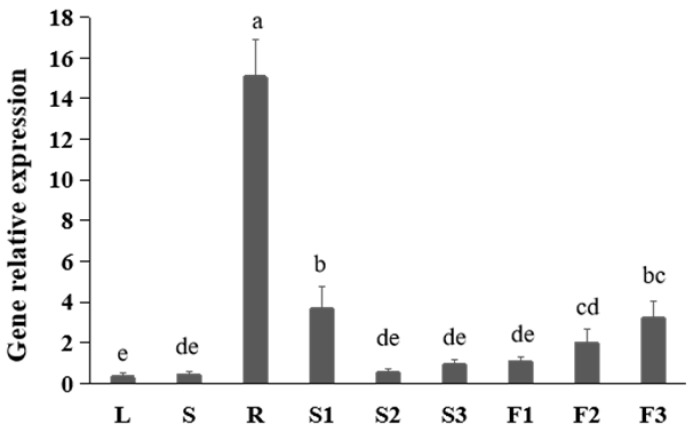
Gene expression analysis of FtNAC10 in different tissues of *F. tataricum*. These tissues are leaves (L), stems (S), roots (R), pre-filling stage seeds (S1), filling stage seeds (S2), fully mature seeds (S3), flower buds (F1), blooming flowers (F2) and near withering flowers (F3), respectively. There are three replicates per sample. Significant differences in gene expression between different tissues were analyzed at level of 0.05. Different lowercase letters indicate significant difference (LSD test, *p* < 0.05).

**Figure 2 ijms-24-16317-f002:**
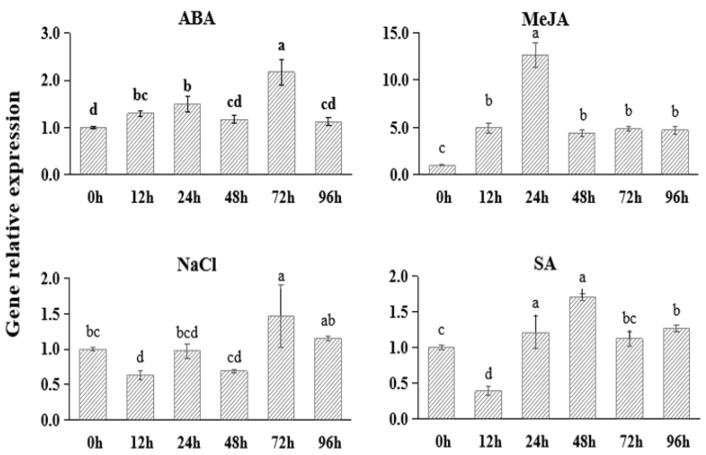
Gene expression analysis of FtNAC10 under different stress treatments in *F. tataricum* seedling. The seedlings were collected at 0, 12, 24, 48, 72 and 96 h after salt (100 µM NaCl), 100 µM abscisic acid (ABA), 100 µM methyl jasmonate (MeJA) and 100 µM salicylic acid (SA) treatment, respectively. Error bars indicate SE (*n* = 3). Different lowercase letters indicate significant difference (LSD test, *p* < 0.05).

**Figure 3 ijms-24-16317-f003:**
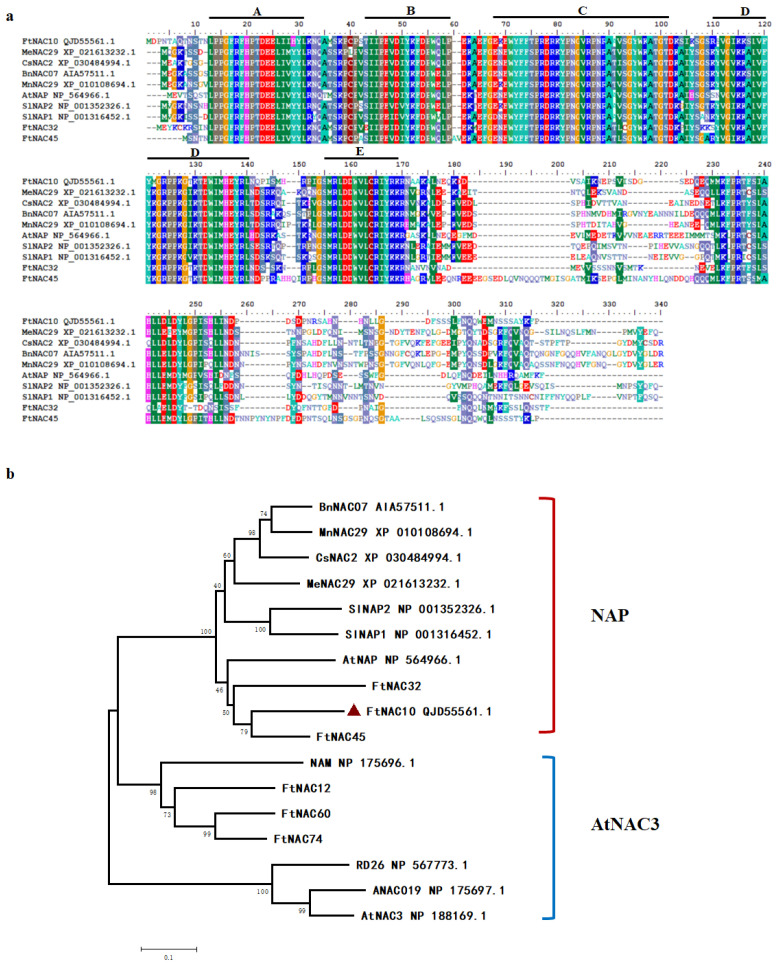
Characterization of the FtNAC10 protein sequences. (**a**) Alignment of FtNAC10 proteins with its homologous NAC proteins from other species: *F*. *tataricum* (FtNAC10, QJD55561.1, FtNAC32 and FtNAC45 [31], *Manihot esculenta* Crantz (MeNAC29, XP_021613232.1), *Cannabis sativa* L (CsNAC2, XP_030484994.1), *Brassica napus* L (BnNAC07, AIA57511.1), *Morus notabilis* C. K. Schneid (MnNAC29, XP_010108694.1), *Arabidopsis thaliana* (AtNAP, NP_564966.1), *Solanum lycopersicum* L (SlNAP2, NP_001352326.1; SlNAP1, NP_001316452.1). A, B, C, D and E represent the conserved domain of NAC gene respectively. A, B, C, D and E represent the conserved domain of NAC gene respectively. (**b**) Phylogenetic tree of FtNAC10 protein with other NAC proteins. AtNAC3 (NP_188169.1), RD26 (NP_567773.1), NAM (NP_175696.1), and ANAC019 (NP_175697.1), FtNAC12 (MW355482) and FtNAC60, were orginal from *A*. *thaliana* and *F*. *tataricum,* respectively. The phylogenetic tree was constructed by MEGA5.2 software. Protein sequences were aligned using ClustalW program in MEGA5.2. The bootstrap was set as 1000 and the phylogenetic tree was analyzed using neighbor-joining method.

**Figure 4 ijms-24-16317-f004:**
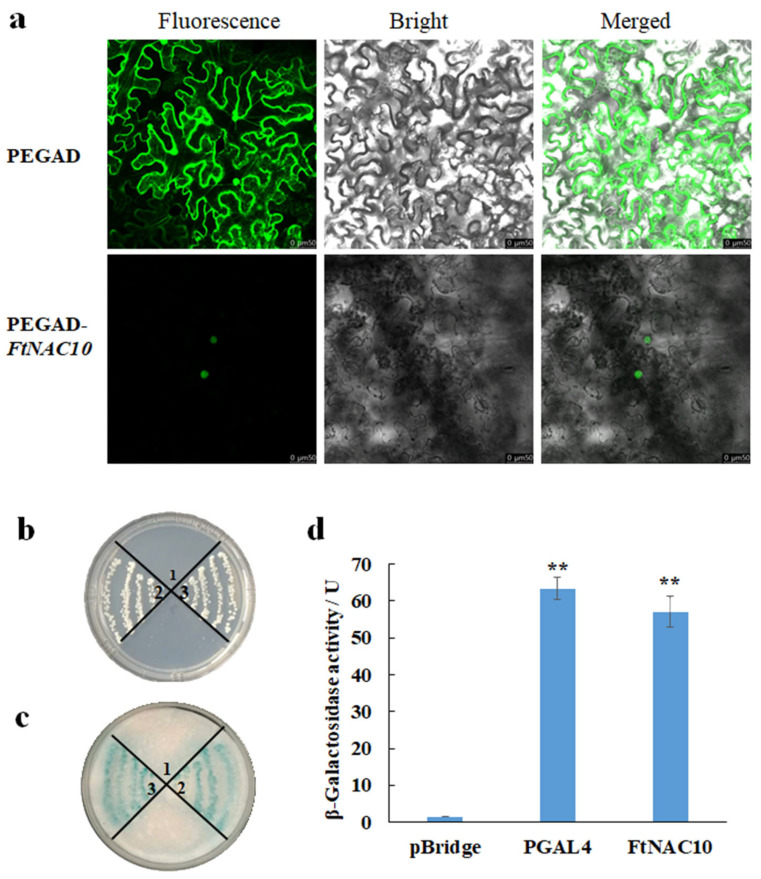
Subcellular localization and transcription activation activity analysis of FtNAC10. 1, 2, and 3 represented pBridge, pBridge-PGAL4, and pBridge-FtNAC10, respectively. (**a**) Subcellular localization of FtNAC10 determined by transforming *N. benthamiana* leaves epidermal cells with PEGAD-GFP and PEGAD-FtNAC10-GFP; (**b**) yeast cultured on SD\-Trp-His; (**c**) The activity analysis results of filter paper. (**d**) Analysis of *β*-galactosidase activity (** *p* < 0.01). Scale bar = 50 µm.

**Figure 5 ijms-24-16317-f005:**
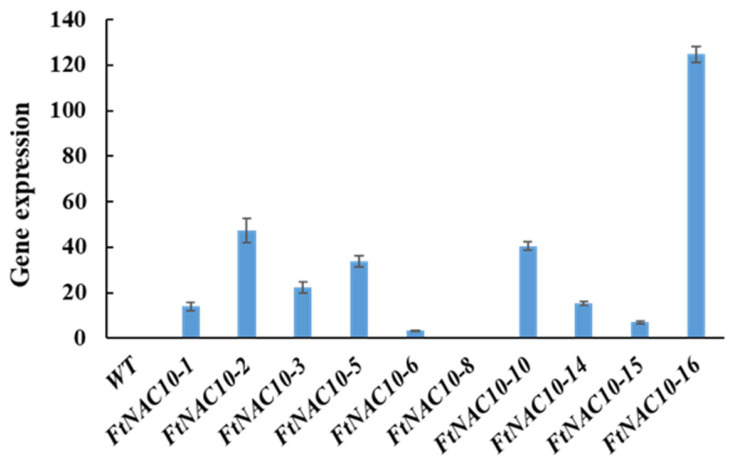
Gene expression analysis of *FtNAC10* in transgenic *A. thaliana*. WT represents control group, 1–16 represent *FtNAC10 A. thaliana* transgenic lines.

**Figure 6 ijms-24-16317-f006:**
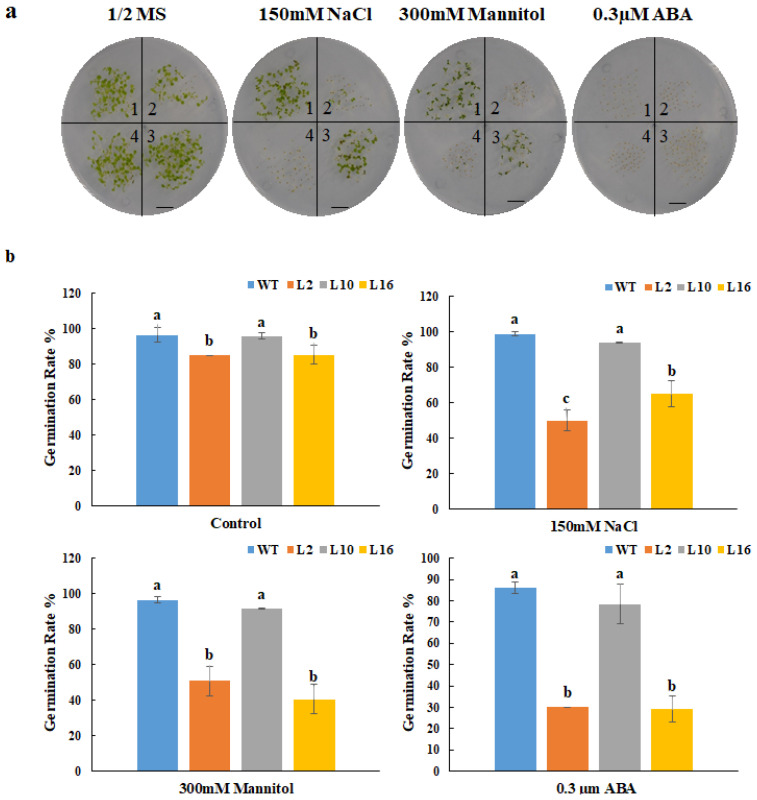
Germination statistical analysis of FtNAC10 transgenic *A. thaliana* lines on stress plate. (**a**) FtNAC10 transgenic *A. thaliana* lines were germinated on plate. The treatments were 1/2 MS (control), 150 mM NaCl, 300 mM Mannitol and 0.3 µM abscisic acid (ABA), respectively. Above figures represent the germination rate of WT and transgenic *A. thaliana* seeds on the 1/2 Murashige and Skoog (MS) plate and with 150 mM NaCl, 300 mM Mannitol and 0.3 µM abscisic acid (ABA), 1, 2, 3 and 4 represent wild type (WT), FtNAC10 transgenic lines L2, L10 and L16, respectively. (**b**) The germination rate of FtNAC10 transgenic lines on stress plate. Significant differences in germination rate between different transgenic lines were analyzed at level of 0.05. Different lowercase letters indicate significant difference (LSD test, *p* < 0.05). Error bars indicate SE (*n* = 3). Scale bar = 1 cm.

**Figure 7 ijms-24-16317-f007:**
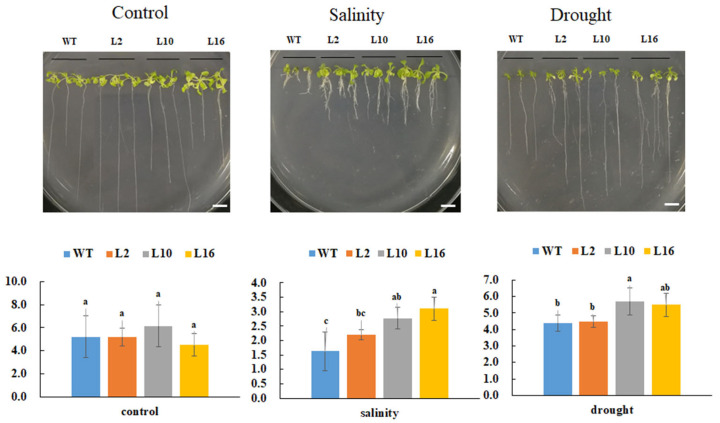
Primary root length analysis of FtNAC10 transgenic *A*. *thaliana* lines under salinity, and drought stress. The stress treatments were salinity (150 mM NaCl), drought (300 mM Mannitol), respectively. WT, L2, L10 and L16 represent wild type, FtNAC10 transgenic lines L2, L10 and L16. All treatments were repeated three times for each line and the data represent means ± SE. Significant differences in root length between different transgenic lines were analyzed at level of 0.05. Different lowercase letters indicate significant difference (LSD test, *p* < 0.05). Scale bar = 1 cm.

## Data Availability

All of the data generated or analyzed during this study are included in this published article and its Appendix A.

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
