# Peer review of "Gene Cloning and Characterization of Transcription Factor FtNAC10 in Tartary Buckwheat (Fagopyrum tataricum (L.) Gaertn.)"

_ijms, 2023, doi:10.3390/ijms242216317_

Round 1
Reviewer 1 Report
Comments and Suggestions for Authors
The manuscript describes the cloning and functional analysis of the transcription factor FtNAC10 in Tartary buckwheat.
The manuscript demonstrates that FtNAC10 functions as a stress-responsive transcriptional activator by gene expression analysis, subcellular localization, and transactivation analysis. Heterologous expression in Arabidopsis reveals that FtNAC10 is involved in seed germination inhibition and stress tolerance. The manuscript will contribute to a better understanding of the functions of the NAC transcription factor in Taratary buckwheat. However, several concerns should be addressed before the acceptance of the manuscript.
1) The manuscript speculates that FtNAC10 regulates ABA biosynthesis (Lines 24 and 304). However, this speculation is not well supported by experimental evidence. Given that ABA induces FtNAC10 expression (Figure 2) and heterologous expression of FtNAC10 enhances seed germination inhibition by ABA (Figure 6), could FtNAC10 work as a downstream signal of ABA? The speculation that FtNAC10 regulates ABA biosynthesis should be revised.
2) Statistical analyses should be performed on Figures 6 and 7.
3) Why is L10 as strong as L2 and L16 in stress tolerance despite its weak seed germination suppression phenotype? What organs were used in the expression analysis in Figure 5? Is it possible that the expression level of FtNAC10 differs between L10 seeds and seedlings?
Author Response
reviewer1: We are very thankful for your time and valuable suggestions on the manuscript. We appreciate the critical comments, and we are glad to respond to the comments highlighted by you. Corrections and detailed information have been highlighted in red in the revised manuscript. 1)The manuscript speculates that FtNAC10 regulates ABA biosynthesis (Lines 24 and 304). However, this speculation is not well supported by experimental evidence. Given that ABA induces FtNAC10 expression (Figure 2) and heterologous expression of FtNAC10 enhances seed germination inhibition by ABA (Figure 6), could FtNAC10 work as a downstream signal of ABA? The speculation that FtNAC10 regulates ABA biosynthesis should be revised. Response: Thank you for your valuable comments. ABA treatment induces FtNAC10 expression in F. tataricum seedling, the over-expression of FtNAC10 in transgenic Arabidopsis thaliana seeds inhibited its germination under stress condition, and the presence of FtNAC10 enhanced root elongation under saline and drought stress. According to phylogenetic analysis and previous reports, our experiments indicate that FtNAC10 may regulate the stress response or development of F. tataricum via ABA-signaling pathway. the sentence has been modified in the manuscript. (Lines 24-26 and 355-356). 2)Statistical analyses should be performed on Figures 6 and 7. Response: Thank you for your suggestion. We have conducted a statistical analysis on Figures 6 and 7, the related part has been revised in the manuscript. 3)Why is L10 as strong as L2 and L16 in stress tolerance despite its weak seed germination suppression phenotype? What organs were used in the expression analysis in Figure 5? Is it possible that the expression level of FtNAC10 differs between L10 seeds and seedlings? Response: Thank you for your valuable feedback. The FtNAC10 gene expression levels (Figure 5) were examined in the seedling leaves of transgenic Arabidopsis T2 generation, while the seed germination experiment was conducted using T3 generation seeds. Both experiments were carried out under different culture conditions and seedling ages. The gene expression level of FtNAC10 may vary during different developmental stages, and there may also be differences in gene expression levels of FtNAC10 between seeds and leaves. We speculate that the expression of FtNAC10 in T3 generation seeds is inadequate, which leads to insufficient sensitivity to these three exogenous treatments during Arabidopsis seed germination.

Reviewer 2 Report
Comments and Suggestions for Authors
Dear authors,
This article is concerning “Gene cloning and function characterization of transcription factor FtNAC10 in Tartary buckwheat, by Jinghuan Li, Xiaohua Li, Caihua Jia and Dahui Liu. As interesting data, I recommend it for an international audience in this journal, however several points have to be considered by the authors, and a major revision is requested.
Please notice that in order to bring a broad audience to this article and to this journal, for specialists and non-specialists, the five major points of my comments (at the beginning) are very important (mandatory…) for a suitable value of the article. Minor points are also enhanced at the end of this review.
I deeply hope to see this good article published soon,
The five major points are:
1-1 The first point embarrassing me is a lot concerns experiments related with figures 6-7 (part of your title “function characterization”, allied with the sentences "Overall, the germination rates for the three transgenic lines were…" in 2.5, "FtNAC10 lines were significantly longer than...” in 2.6), since actually two kinds of diagrams-comparisons have to be provided to state clearly about the significance or not of the experiments: 1/between germinations and roots for the same test presented on one diagram as provided in the present submission; 2/ within each line (WT, L2 10 16), i.e. four diagramsX2 (germinations and roots)) gathering on each of the eight diagrams all the experiments on germinations (control, NaCl, Mannitol, ABA) or roots (control, salinity, drought). Then, as means and standard deviation are too weak evaluations as in the present text, apply a classical T test student or equivalent (as you have 30 measurements for each experiment (?)) for each diagram of 1/ and 2/ and provide clearly the p values (I cannot see them) to state efficiently about the significance or not of the differences, which will increase a lot the value of your article. Just looking at the diagrams provided in this submission, I suspect that some comparisons are not significantly different at all and just show (at maximum) a tendency...).
2-2 Linked to the first point, since you found “only” a correlation (?) between germinations-roots and the effect of NaCl and etc…, but no “chain of causation” (which is very different and much more complex to show or not, but allowing the notion of function), you cannot write in your abstract "This study provides a theoretical basis for…", at the end of the introduction, and in the title "function characterizations”, and in 2.6 "to investigate the function..." as you pretend. There are many correlations within plants world, but only very few demonstrations of chains of causation and functions...).
3- 3 Restrict the results part to only results (values etc.) and remove all remarks and sentences which belong actually to the discussion part (e.g. in 2.3, the last sentences are part of the discussion? the last sentence of 2.5 (these results indicated...") is actually part of the discussion? in the discussion, in order to connect results and discussion parts, indicate some values of your own results-experiments.
4-4 Although interesting, the phylogeny remark needs much more discussion and references to have a real value.
5- References already taken in account by the authors are of interest, however checking briefly in word of science WOS and scilit (from mdpi) with some key-words of this manuscript, other references appear, and they should be updated and used (if relevant…) in order to provide a larger view of these interesting researches. Among these are the followings (check all others):
[1-13]
1. Aubert, L.; Quinet, M. Comparison of Heat and Drought Stress Responses among Twelve Tartary Buckwheat (Fagopyrum tataricum) Varieties. Plants-Basel 2022, 11, doi:10.3390/plants11111517.
2. Deng, R.Y.; Zhao, H.X.; Xiao, Y.H.; Huang, Y.J.; Yao, P.F.; Lei, Y.L.; Li, C.L.; Chen, H.; Wu, Q. Cloning, Characterization, and Expression Analysis of Eight Stress-Related NAC Genes in Tartary Buckwheat. Crop Sci 2019, 59, 266-279, doi:10.2135/cropsci2018.06.0368.
3. Huang, J.; Chen, Q.J.; Rong, Y.P.; Tang, B.; Zhu, L.W.; Ren, R.R.; Shi, T.X.; Chen, Q.F. Transcriptome analysis revealed gene regulatory network involved in PEG-induced drought stress in Tartary buckwheat (Fagopyrum Tararicum). Peerj 2021, 9, doi:10.7717/peerj.11136.
4. Huang, J.; Ren, R.R.; Rong, Y.P.; Tang, B.; Deng, J.; Chen, Q.F.; Shi, T.X. Identification, Expression, and Functional Study of Seven NAC Transcription Factor Genes Involved in Stress Response in Tartary Buckwheat (Fagopyrum tataricum (L.) Gaertn.). Agronomy-Basel 2022, 12, doi:10.3390/agronomy12040849.
5. Li, Q.; Wu, Q.; Wang, A.H.; Lv, B.B.; Dong, Q.X.; Yao, Y.J.; Wu, Q.; Zhao, H.X.; Li, C.L.; Chen, H.; et al. Tartary buckwheat transcription factor FtbZIP83 improves the drought/salt tolerance of Arabidopsis via an ABA-mediated pathway. Plant Physiology and Biochemistry 2019, 144, 312-323, doi:10.1016/j.plaphy.2019.10.003.
6. Li, Q.; Zhao, H.X.; Wang, X.L.; Kang, J.Y.; Lv, B.B.; Dong, Q.X.; Li, C.L.; Chen, H.; Wu, Q. Tartary Buckwheat Transcription Factor FtbZIP5, Regulated by FtSnRK2.6, Can Improve Salt/Drought Resistance in Transgenic Arabidopsis. Int J Mol Sci 2020, 21, doi:10.3390/ijms21031123.
7. Liu, M.Y.; Ma, Z.T.; Sun, W.J.; Huang, L.; Wu, Q.; Tang, Z.Z.; Bu, T.L.; Li, C.L.; Chen, H. Genome-wide analysis of the NAC transcription factor family in Tartary buckwheat (Fagopyrum tataricum). Bmc Genomics 2019, 20, doi:10.1186/s12864-019-5500-0.
8. Liu, M.Y.; Wen, Y.D.; Sun, W.J.; Ma, Z.T.; Huang, L.; Wu, Q.; Tang, Z.Z.; Bu, T.L.; Li, C.L.; Chen, H. Genome-wide identification, phylogeny, evolutionary expansion and expression analyses of bZIP transcription factor family in tartaty buckwheat. Bmc Genomics 2019, 20, doi:10.1186/s12864-019-5882-z.
9. Lv, B.B.; Wu, Q.; Wang, A.H.; Li, Q.; Dong, Q.X.; Yang, J.J.; Zhao, H.X.; Wang, X.L.; Chen, H.; Li, C.L. A WRKY transcription factor, FtWRKY46, from Tartary buckwheat improves salt tolerance in transgenic Arabidopsis thaliana. Plant Physiology and Biochemistry 2020, 147, 43-53, doi:10.1016/j.plaphy.2019.12.004.
10. Song, J.N.; Wang, Y.Q.; Yang, H.B. MicroRNA sequencing reveals the salt responses in four Tartary buckwheat cultivars. Theoretical and Experimental Plant Physiology 2022, 34, 517-536, doi:10.1007/s40626-022-00262-0.
11. Xiao, S.Y.; Liu, Y.; Wang, A.H.; Liu, Y.D.; Li, X.Y.; Liu, Z.B.; Li, X.F.; Yang, Y.; Wang, J.M. The response of tartary buckwheat and 19 bZIP genes to abscisic acid (ABA). Molecular Biology Reports 2021, 48, 4341-4350, doi:10.1007/s11033-021-06449-z.
12. Yao, P.F.; Li, C.L.; Zhao, X.R.; Li, M.F.; Zhao, H.X.; Guo, J.Y.; Cai, Y.; Chen, H.; Wu, Q. Overexpression of a Tartary Buckwheat Gene, FtbHLH3, Enhances Drought/Oxidative Stress Tolerance in Transgenic Arabidopsis. Frontiers in Plant Science 2017, 8, doi:10.3389/fpls.2017.00625.
13. Zhao, J.L.; Wu, Q.; Wu, H.L.; Wang, A.H.; Wang, X.L.; Li, C.L.; Zhao, H.X.; Wu, Q. FtNAC31, a Tartary buckwheat NAC transcription factor, enhances salt and drought tolerance in transgenic Arabidopsis. Plant Physiology and Biochemistry 2022, 191, 20-33, doi:10.1016/j.plaphy.2022.09.016.
Minor points are
1 As I am involved in plant taxonomy I am very sensible to correct plant taxa names which make their homogeneity and precision at the international level. In the present text this is too heterogeneous (English names for some taxa (rice, litchi… see all others), latin for others with only genus name; for instance Arabidopsis is a genus name and it has no precise taxonomical meaning at all, it has to be followed by species name (there are actually several Arabidopsis taxa), then once it is written correctly in full letters you can write further in the text “A. plus species name plus author’s name”, even in titles and captions of figures. So write all plants with (English names if you wish and) latin names in brackets and in italics, followed by the names of the author(s) at least the first time they appear in the text (from the beginning of the introduction, see also for all other parts as there are numerous mistakes). Use international Plant Names Index (IPNI) https://www.ipni.org/, or equivalent; for instance just for cultivars, if there is no author’s name(s), put the reference where this name appears firstly in the literature.
2 for figure 1, in order to be read rapidly indicate in the caption what are the abbreviations used (in brackets, just after each word in full letters);
3 for figure 2, indicate in the caption the meaning in full letters of ABA, MJA, SA;
4 for figure 3, the amino-acids (?) in colors are almost invisible, as well as the letters in grey in the first left column; moreover indicate in the caption the meaning of A, B, C, D, E above the sequences; the phylogenetic tree (b) has 8 entries while there are 10 in a?
5 the caption of figure 4 is not at all understandable: what is the relationship between a, b, c, d? What mean exactly the colored rows (white and green) in 2 and 3 of b-c? in a, “Bright” means with no fluorescence ? N.benthamiana should be in italics and there is a space between the genus abbreviation and the species name;
6 for figure 6, the relationship between the upper and lower parts is not clear at all (is 1/2MS the control? 1 2 3 4 in the upper part should be indicated also just above or below WT L2 10 16?); indicate in the caption in full letters the meaning of ABA, MS; moreover Arabidopsis should be in italics in the title and in the caption;
7 figure 7 is not clear at all: salinity test corresponds to NaCl and drought to mannitol? The roots measurements are indicated with standard deviation?
8 in the discussion, last paragraph, indicate and detail which "Numerous studies have found...”;
9 in the conclusion, indicate in full letters the meaning of TF.
Comments on the Quality of English Language
I am not an english native speaker. Actually it is fully understandable however in some places the english grammar needs to be checked again more carefully
Author Response
Reviewer2
We are very thankful for your time and valuable suggestions on the manuscript. We appreciate the critical comments, and we are glad to respond to the comments highlighted by you. Corrections and detailed information have been highlighted in red in the revised manuscript.
1-1 The first point embarrassing me is a lot concerns experiments related with figures 6-7 (part of your title “function characterization”, allied with the sentences "Overall, the germination rates for the three transgenic lines were…" in 2.5, "FtNAC10 lines were significantly longer than...” in 2.6), since actually two kinds of diagrams-comparisons have to be provided to state clearly about the significance or not of the experiments: 1/between germinations and roots for the same test presented on one diagram as provided in the present submission; 2/ within each line (WT, L2 10 16), i.e. four diagramsX2 (germinations and roots)) gathering on each of the eight diagrams all the experiments on germinations (control, NaCl, Mannitol, ABA) or roots (control, salinity, drought). Then, as means and standard deviation are too weak evaluations as in the present text, apply a classical T test student or equivalent (as you have 30 measurements for each experiment (?)) for each diagram of 1/ and 2/ and provide clearly the p values (I cannot see them) to state efficiently about the significance or not of the differences, which will increase a lot the value of your article. Just looking at the diagrams provided in this submission, I suspect that some comparisons are not significantly different at all and just show (at maximum) a tendency...).
Response: Thank you for your valuable suggestion. We have clearly recognized this problem, and we have revised Figure 6-7 in the manuscript. The gene expression level of FtNAC10 may vary during different developmental stages, the significant differences were compared between WT and transgenic lines (p < 0.05) during generation, the transgenic L2 shown significant differences at 2 days or 4 days during generation. we speculate that the expression of FtNAC10 in T3 generation seeds is inadequate, which leads to insufficient sensitivity to these exogenous treatments during Arabidopsis seed germination.
2-2 Linked to the first point, since you found “only” a correlation (?) between germinations-roots and the effect of NaCl and etc…, but no “chain of causation” (which is very different and much more complex to show or not, but allowing the notion of function), you cannot write in your abstract "This study provides a theoretical basis for…", at the end of the introduction, and in the title "function characterizations”, and in 2.6 "to investigate the function..." as you pretend. There are many correlations within plants world, but only very few demonstrations of chains of causation and functions...).
Response: thank you for your valuable suggestion. We have revised this part according to the reviewer’s suggestion in abstract, 2.6, (Line 26, 185) and in the title.
- 3 Restrict the results part to only results (values etc.) and remove all remarks and sentences which belong actually to the discussion part (e.g. in 2.3, the last sentences are part of the discussion? the last sentence of 2.5 (these results indicated...") is actually part of the discussion? in the discussion, in order to connect results and discussion parts, indicate some values of your own results-experiments.
Response: Thank you for your valuable suggestion. We have revised according to your suggestion in results (e.g 2.3, 2.5) and discussion parts (Lines 241-252).
- 4- Although interesting, the phylogeny remark needs much more discussion andreferences to have a real value.
Response: Thank you for your valuable suggestion. We have revised with more discussion and references according to your suggestion (Lines 201-207,219-223,226-228,233-236).
5- References already taken in account by the authors are of interest, however checking briefly in word of science WOS and scilit (from mdpi) with some key-words of this manuscript, other references appear, and they should be updated and used (if relevant…) in order to provide a larger view of these interesting researches. Among these are the followings (check all others):
Response: Thank you for your valuable suggestion. We have revised in text and references according to your suggestion (Lines 59, 199-207, 210-212,219-223,226-228,233-236, references 31, 33-34, 42-46).
Minor points are
1 As I am involved in plant taxonomy I am very sensible to correct plant taxa names which make their homogeneity and precision at the international level. In the present text this is too heterogeneous (English names for some taxa (rice, litchi… see all others), latin for others with only genus name; for instance Arabidopsis is a genus name and it has no precise taxonomical meaning at all, it has to be followed by species name (there are actually several Arabidopsis taxa), then once it is written correctly in full letters you can write further in the text “A. plus species name plus author’s name”, even in titles and captions of figures. So write all plants with (English names if you wish and) latin names in brackets and in italics, followed by the names of the author(s) at least the first time they appear in the text (from the beginning of the introduction, see also for all other parts as there are numerous mistakes). Use international Plant Names Index (IPNI) https://www.ipni.org/, or equivalent; for instance just for cultivars, if there is no author’s name(s), put the reference where this name appears firstly in the literature.
Response: Thank you for your suggestion. We have revised the plant names in this paper (Lines 3,15,22,49,53,120-123 etc.).
2 for figure 1, in order to be read rapidly indicate in the caption what are the abbreviations used (in brackets, just after each word in full letters);
Response: Thank you for your comments. We have revised it carefully accroding to your suggestion (Lines 96-98).
3 for figure 2, indicate in the caption the meaning in full letters of ABA, MJA, SA;
Response: Thank you for your comment. We have revised these information for figure 2 (Lines 102-104).
4 for figure 3, the amino-acids (?) in colors are almost invisible, as well as the letters in grey in the first left column; moreover indicate in the caption the meaning of A, B, C, D, E above the sequences; the phylogenetic tree (b) has 8 entries while there are 10 in a?
Response: Thank you for your comment. We have revised figure 3 and legend (lines117-127). We added more NAC proteins in figure 3b in order to better present the phylogenetic relationships and functional grouping with other NAC proteins.
5 the caption of figure 4 is not at all understandable: what is the relationship between a, b, c, d? What mean exactly the colored rows (white and green) in 2 and 3 of b-c? in a, “Bright” means with no fluorescence ? N.benthamiana should be in italics and there is a space between the genus abbreviation and the species name;
Response: Thank you for your comment. Subcellular localization (a) and transcription activation activity analysis (b,c,d) are belong to basic analysis for molecular biology characteristics of TFs, Subcellular localization (a) is to show the subcellular localization of the protein, which can be marked with GFP tag, and the fluorescence signal of GFP was observed and compared by confocal microscope with different lighting methods. Finally, fluorescent images of GFP , bright light images and the merged images were listed to show the results in text. The “bright” field allows the light beam to pass through the specimen directly into the objective lens, resulting in a bright view. We detected the transcription activity of NAC in yeast by β-galactosidase colony-lift filter assay according to Yeast Protocol Handbook (Clontech Laboratories, Inc.), the colored rows changed from white (b) to green (c) means it exhibited transcriptional activity. Figure 4 d is another method to measure the β-galactosidase activity to show whether FtNAC10 exhibits transcriptional activation activity or not. We have fomatted “N. benthamiana” in line 153.
6 for figure 6, the relationship between the upper and lower parts is not clear at all (is 1/2MS the control? 1 2 3 4 in the upper part should be indicated also just above or below WT L2 10 16?); indicate in the caption in full letters the meaning of ABA, MS; moreover Arabidopsis should be in italics in the title and in the caption;
Response: Thank you for your comments. We have revised these according to your suggestions. 1/2 MS is the control without any stress treatments.(Figure 6 and Lines 175-181)
7 figure 7 is not clear at all: salinity test corresponds to NaCl and drought to mannitol? The roots measurements are indicated with standard deviation?
Response: Thank you for your comments. We have revised Figure 7 and legend according to two reviewers’ suggestions (Figure 7 and Lines 193-197).
8 in the discussion, last paragraph, indicate and detail which "Numerous
studies have found...”;
Response: Thank you for your comments. We have revised this part with references [12, 23, 24, 41, 43].(Lines 262-263).
9 in the conclusion, indicate in full letters the meaning of TF.
Response: Thank you for your comments. We have revised it in text. (Line 353 )
